# A Heterogeneous Sample of a Spanish Tinnitus Cohort

**DOI:** 10.3390/brainsci13040652

**Published:** 2023-04-12

**Authors:** María Cuesta, Pedro Cobo

**Affiliations:** Institute for Physical and Information Technologies (ITEFI), Spanish National Research Council (CSIC), 28006 Madrid, Spain; m.cuesta@csic.es

**Keywords:** tinnitus, heterogeneity, hearing loss, tinnitus severity, gender, age

## Abstract

Taking into account heterogeneity has been highly recommended in tinnitus studies both to disentangle all diverse factors that can contribute to their complexity and to design personalized treatments. To this aim, a heterogeneous sample of 270 tinnitus subjects is analyzed considering the gender (male/female), hearing condition (hearing-impaired/normal-hearing), and tinnitus severity (compensated/decompensated) subgroups. Two categorical variables (tinnitus laterality and tinnitus sound type) and four quantitative variables (average auditory threshold, age of tinnitus onset, tinnitus frequency, and tinnitus severity) are used. The percentages (for categorical variables) and mean values (for quantitative variables) of the whole sample are compared with these of each subgroup. Furthermore, correlational and hypothesis testing is applied to calculate the correlation coefficients and statistical significance, respectively. The results show that the male and female subgroups contrast in the sound type and frequency of their tinnitus, hearing-impaired and normal-hearing individuals differ, in addition, in their average auditory threshold, and the compensated/decompensated tinnitus subgroup provides significantly distinct values in tinnitus laterality and tinnitus sound.

## 1. Introduction

Tinnitus is the perception of a sound that has not been generated by any external source. When this sound has its origin in a source in the body (e.g., muscle contractions or blood flow), tinnitus is named objective. On the other hand, if this sound arises as an aberrant plastic compensation mechanism in the neural auditory system, it is called subjective [1]. Alternative names for subjective/objective tinnitus are primary/secondary tinnitus [1]. Other authors refer to secondary/primary tinnitus depending on whether it can or cannot be associated with any organic cause [2]. Other classifications of tinnitus take into account its duration, distinguishing between acute and chronic, depending on whether its duration is lesser or larger than 3–6 months [1].

Tinnitus is a highly heterogeneous disorder [3], having multiple risk factors (hearing loss, vestibular disorders, emotional disorders, vascular risk factors, neck and temporomandibular dysfunction, head trauma, etc.) [4] and a variety of possible mechanisms (hyperactivity, hypersynchrony, cortical tonotopic map reorganization [5], or quantum tunneling of calcium, potassium, and sodium ions through the closed voltage-gated channels [6]).

In general, two subjects having tinnitus triggered by the same cause, or produced by the same mechanism, will likely have different responses, in terms of perception (laterality, sound type and pitch, and loudness) and severity, since tinnitus development is mediated by demographic and socioeconomic factors (e.g., age, gender, and education) [7], personality and psychological traits (e.g., obsessive, ruminant, neurotic, etc.) [8,9], and mental health condition (e.g., stress, anxiety, and depression) [10].

In view of this, innovative research aimed to better understand tinnitus heterogeneity has been proposed [3]. Understanding phenotypic differences in tinnitus patients could help to disentangle distinct tinnitus mechanisms and to design more effective customized tinnitus treatments [11]. Understanding the intrinsic heterogeneity of tinnitus individuals and identifying potential subgroups can be important in revealing other influences on tinnitus [12,13].

Although hearing loss is a main risk factor of tinnitus [14], many authors have found that a percentage of them (between one-fifth and one-third) have apparently a normal audiogram [15,16,17]. According to the primary/secondary classification, tinnitus related to hearing loss belongs to the primary tinnitus group [2]. Some authors have already studied the relationship between hearing levels and tinnitus-related distress, taking into consideration gender-based differences [18]. The generally used audiometry in everyday practice has the limitation that measures the 125–8000 Hz frequency spectrum. Even though an audiometry with normal hearing levels does not exclude high-frequency hearing losses [19], hidden hearing losses [20], or hearing losses at other non-measured frequencies [21], it is tempting to carry out a first subgrouping between subjects with normal hearing and subjects who are hearing-impaired, according to their pure tone audiometry.

The sex/gender of tinnitus subjects can also be a distinctive factor [18,22,23]. While the tinnitus-related distress of male patients seems to be more linked to personality, this distress in female patients is more connected with emotions [18]. Significantly more males report tinnitus [22], although females used to report a higher emotional reaction to their tinnitus than males [18]. Other studies found gender differences regarding the laterality or severity of tinnitus [23]. Thus, a gender-based study of tinnitus subjects is also highly appealing.

The tinnitus-related distress should be considered in the design of customized treatments for tinnitus. Hiller et al. proposed to classify tinnitus patients in the compensated (slight and moderate distress) and decompensated (severe and catastrophic distress) subgroups depending on their tinnitus severity outcome [8] as assessed by the Tinnitus Questionnaire. Systematic differences have been reported in the personality factors of decompensated and compensated tinnitus patients [9]. Therefore, the subdivision of tinnitus patients into the compensated and decompensated subgroups could also provide relevant outcomes.

The main aim of this article is to provide a stratified analysis of the demographic, audiometric, and tinnitus-related characteristics of a cohort of 270 Spanish tinnitus subjects. The analysis result will be carried out for the whole sample as well as for the gender (male/female), hearing condition (normal-hearing/hearing-impaired), and tinnitus-related distress (compensated/decompensated) subgroups.

## 2. Materials and Methods

### 2.1. Participants

For this study, which ran from January 2018 to December 2022, 270 tinnitus participants were recruited. The study was approved by the CSIC Ethics Committee. All participants signed an informed consent form. Most of them joined this study after having visited several other specialists, such as otolaryngologists, audiologists, neurologists, psychologists, psychiatrists, physiotherapists, etc.

The mean and standard-deviation (SD) age (for all participants and by male/female gender) are summarized in Table 1. Men were slightly older (two years) than women.

### 2.2. Audiometric Assessment

Participants were required to provide their hearing levels (HL) measured in any external clinic at eleven frequencies from 125 Hz to 8 kHz. Figure 1 shows the mean HL of the 270 participants in left and right ears. The shaded area around the curves outlines the confidence intervals (±1.96 × SD/sqrt(N)), with SD being the standard deviation and N the number of subjects. Participants exhibited slightly higher HL at the left ear. Their mean levels below 2 kHz were normal while, at higher frequencies, they displayed mild-to-moderate HL.

When participants were sorted by gender, the mean HL for male and female are shown in Figure 2. Again, in both cases, the left ear HL was higher than the right ear HL. On the other hand, the mean HL curve of males exhibited a steeper slope at high frequencies than females. At 8 kHz, for instance, the mean was a 7 dB-larger HL for males. The average HL at high frequencies (3–8 kHz) for males and females were 39.1 and 31.6 dB, respectively. A *t*-test comparing these values showed that this difference was statistically significant at *p* < 0.05.

Participants were also categorized according to their hearing condition. Average auditory thresholds (AAT) at all frequencies and at low (AAT_LF_) and high (AAT_HF_) frequencies were used, to this aim. Specifically, AAT is the mean HL at the eleven measured frequencies, AAT_LF_ is the mean HL at the seven frequencies between 125 Hz and 2 kHz, and AAT_HF_ is the mean HL at the four frequencies from 3 to 8 kHz.

Participants were sorted in hearing-impaired (HI) and normal-hearing (NH) subgroups using Algorithm 1:
**Algorithm 1:** Division of different listening participants1. If any HL(*f_i_*) ≥ 40 dB2. or AAT ≥ 30 dB3. or $\left|{AAT}_{HF}-{AAT}_{LF}\right|\geq 17dB$4.   participant is included in the HI subgroup5. else6.   participant is incorporated to the NH subgroup
where *f_i_* is any of the eleven frequencies of the measured audiometry. Notice that the first condition detects mainly scotomas, the second identifies essentially conductive hearing losses, while the third finds principally sensorineural hearing losses.

This algorithm detected 201 (74%) HI and 69 (26%) NH participants. Figure 3 displays the mean HL curves of these groups. As expected, the mean HL of the NH subgroup was lesser than 20 dB. The mean HL of the HI participants showed a steeped loss above 2 kHz, with slightly higher loss in the left ear as compared to that of the right ear.

### 2.3. Tinnitus Assessment

Several tinnitus characteristics of participants, such as their tinnitus location (left ear, right ear, or bilateral), tinnitus sound (tonal, ringing, or hissing), tinnitus duration (time after tinnitus onset), tinnitus frequency, tinnitus severity, and possible cause or origin of their tinnitus were evaluated just after being accepted in the study. Specifically, the tinnitus location, duration, and possible cause of participants were figured out through the case history.

Many of the participants reported multiple possible causes of their tinnitus onset. The predominant circumstances were emotional troubles (stress, anxiety, depression, or obsessive-compulsive syndrome) and hearing losses (conductive, sensorineural, or sudden). Other causes were noise (overexposure or trauma), hearing troubles (otitis or tubaritis), middle ear surgery (e.g., cholesteatoma or otosclerosis), vestibular disorders (e.g., Ménière’s disease), and other central causes (e.g., Chiari malformation, meningioma, or hydrocephaly).

The frequency and type of sound were found by a matching procedure. Participants were exposed to a band noise whose central frequency and bandwidth could be adjusted. Depending on the noise bandwidth, the tinnitus sound was classified as tonal, when the bandwidth was lesser than 1% of the central frequency; ringing, for a bandwidth between 1% and 9% of the central frequency; and hissing, for wideband larger than 10% of the central frequency. First, the subject was asked to describe their tinnitus sound. Most of them referred to tonal sound as being like a whistle, ringing sound like a cricket or cicada, and hissing sound like a badly tuned radio or the noise of steam from a pressure cooker.

The degree of tinnitus severity was estimated through the validated Spanish version of the Tinnitus Handicap Inventory (THI) [24], which measures 3 dimensions of symptoms (i.e., functional, emotional, and catastrophic). The THI scores the tinnitus severity between 0 and 100 [25,26] and classifies it into five grades [27]: slight (THI ≤ 16), mild (18 ≤ THI ≤ 36), moderate (38 ≤ THI ≤ 56), severe (58 ≤ THI ≤ 76), and catastrophic (THI ≥ 78).

Hiller et al. [8] used another questionnaire, a German version of the Tinnitus Questionnaire (TQ), to sort the tinnitus of patients. The TQ ranges from 0 to 84 and also distinguishes between slight to moderate tinnitus (0 ≤ TQ ≤ 46) and severe to catastrophic tinnitus (47 ≤ TQ ≤ 84) [9]. Tinnitus-related distress is named ‘compensated’ for slight and moderate grades and ‘decompensated’ for severe and catastrophic levels. Although the THI and the TQ have different score ranges, the same distress degrees of THI can be used to sort tinnitus patients in compensated/decompensated subgroups. Therefore, 165 participants (61%) were included in the compensated subgroup (THI ≤ 57), while 105 (39%) were incorporated into the decompensated subgroup (THI ≥ 57).

### 2.4. Statistical Analysis

All data were analyzed using MATLAB (The Mathworks Inc, Natick, MA. USA). Descriptive statistics was applied to analyze categorical variables (tinnitus sound type and tinnitus location). Quantitative variables (age, tinnitus duration, tinnitus frequency, AAT, and THI) were analyzed by means of descriptive and correlational statistics. In addition to correlation coefficients, significance levels (*p*) were provided. Hypothesis testing (*t*-test) was applied to assess statistical significance of differences between mean values of these variables at *p* < 0.05 significance level.

## 3. Results

Results will be provided for all (270) participants and for three subgroups: gender (male/female), hearing condition (HI/NH), and tinnitus severity (compensated/decompensated).

### 3.1. Tinnitus Location and Tinnitus Sound Type

The number and percentage of the categorical variables ‘tinnitus location’ and ‘tinnitus sound’ are summarized in Table 2. Tinnitus was located in both ears for almost half of the participants, in the left ear for roughly one-third of the subjects, and in the right ear for approximately one-sixth of the patients. These percentages varied only slightly when analyzed for the subgroups.

### 3.2. Age and Tinnitus Duration

The mean and standard deviation (SD) of the age and tinnitus duration of the participants are summarized in Table 3. The mean age was slightly lower (2 years) for women than for men. The HI participants were older (7 years) than the NH subjects. Moreover, the patients with compensated tinnitus were slightly older (2.5 years) than those with decompensated tinnitus.

The risk age for developing tinnitus (the mean age of tinnitus onset) has been assessed as the mean age minus the mean tinnitus duration of a participant. Curiously, this risk age was around 45 years, except for the NH subgroup which was 41 years.

### 3.3. Average Auditory Thresholds, Tinnitus Frequency, and Tinnitus Severity

The mean and SD of the quantitative variables, average auditory thresholds (AAT), tinnitus frequency, and tinnitus severity (THI), are summarized in Table 4. For the average auditory thresholds, the mean and SD values were provided for the whole frequency range as well as for low and high frequencies. For the AAT variable, no significant differences were found for all the participants and the gender and tinnitus severity subgroups. However, important changes were observed for the male/female subgroups in the AAT_LF_ and AAT_HF_ variables. Specifically, men had slightly lower values of AAT_LF_ (3 dB) but notably higher values of AAT_HF_ (7 dB) than women. These differences were statistically significant at *p* < 0.05. The mean tinnitus frequency was significantly higher (1380 Hz) (*p* < 0.05) in men than women, lower (417 Hz) in HI than in NH, but roughly the same in compensated/decompensated subjects. On the other hand, no significant differences were found in the mean tinnitus severity of the male/female and HI/NH subgroups.

### 3.4. Audiometric Characteristics/Tinnitus Features Correlations

Correlational analysis was applied for the quantitative variables AAT, tinnitus frequency, and THI for all participants, as well as for those included in the three subgroups. AAT was considered the independent variable while tinnitus frequency and tinnitus severity (THI) were the dependent variables. Table 5 summarizes the correlation coefficients of each pair of variables. The tinnitus frequency and the AAT were negatively correlated (tinnitus frequency decreases with increasing AAT), with good significance levels, except for the female and NH subgroups. As an example, Figure 4 shows the scatter plot of tinnitus frequency versus AAT for the decompensated subgroup.

The tinnitus severity (THI) and the AAT were positively correlated (THI increases with increasing AAT), also with good significance levels, again except for NH subgroup. As an example, Figure 5 shows the scatter plot of tinnitus severity versus AAT for the decompensated subgroup.

## 4. Discussion

More men (66%) than women (34%) were recruited in this study. This means that approximately twice the number of men than women asked for help with their tinnitus. Similar percentages have been reported in other studies with similar percentages (60% versus 40% in [13], 66% versus 34% in [28], and 61% versus 39% in [22]). However, although significantly more males report tinnitus, females seem to progress to greater emotional distress [22].

Table 6 summarizes the mean values of the categorical (tinnitus laterality and tinnitus sound) and quantitative (age of tinnitus onset, AAT, tinnitus frequency, and THI) variables for all participants, and the relative variation of those variables for each subgroup with respect to all. A percent variation lesser than ±2% for the categorical variables, values within ±2 dB of the AAT, variations of tinnitus frequency lesser than ±200 Hz, and variations of the THI score smaller than ±2 are considered non-significant and marked with the symbol ≈. Significantly higher and lower values are marked with the symbols ↑ and ↓, respectively.

Roughly half of participants located their tinnitus bilaterally, one-third in the left ear, and one-sixth in the right ear. The percentages for bilateral/unilateral are very similar to those reported by Van den Berge et al. [13]. Sharma et al. [22] also found a higher prevalence of tinnitus laterality towards the left ear and suggested a certain hearing asymmetry as a possible explanation of this bias. We did not find significant differences in this distribution of tinnitus laterality for the gender (male/female) and hearing condition (HI/NH) subgroups, but we did for the tinnitus severity subgroups. Specifically, subjects with decompensated tinnitus locate their sound more bilaterally and less towards the left ear.

The more frequent tinnitus sound type in our cohort was hissing (38%), followed by tonal (33%) and ringing (29%). A comparison with other studies is difficult, as other authors use different sound denominations. Van den Berge et al. [13], for instance, described tinnitus sound as tonal (44.5%), noise (44.1%), and other (11.4%). In our analysis, the percentage of subjects perceiving a hissing sound decreased for the female, NH, and compensated subgroups, and increased for the HI and decompensated subgroups. More participants in the female, NH, and compensated subgroups, but less subjects with decompensated tinnitus, perceived a tone. On the other hand, less male subjects and more subjects with NH noticed a ringing.

The average auditory threshold, both for the whole frequency range (AAT) and for low (AAT_LF_) and high frequency (AAT_HF_) ranges, were 25 dB, 18 dB, and 37 dB, respectively. Aazh and Salvi [29] provided the same AAT for a sample of 445 patients in the UK, Brueggemann et al. [12] reported a slightly lower AAT (23 dB) in their cohort of 531 patients, while Van den Berge et al. [13] reported a slightly higher AAT (29.7 dB) in their study on 1764 subjects in The Netherlands. In our cohort, the three variables are the same for the gender and tinnitus severity subgroups (for hearing condition, the subgroups are different, as sorting patients in the HI and NH subgroups is based in AAT).

The mean age (51.3 years) and tinnitus duration (6.5 years) of the participants in our study are similar to those of other authors. For instance, the mean age of the cohort of Brueggemann et al. [12] was 49 years, while the mean age and mean tinnitus duration of participants in the study of Van den Berge et al. [13] were 53.6 years and 6.8 years, respectively. If the mean age of tinnitus onset is assessed as the difference between the mean age and the mean tinnitus duration, our participants made their tinnitus debut at 44.8 years. This date was 46.8 in the cohort of Van den Berge et al. [13]. The risk date of this cohort to the tinnitus debut remains almost the same when the data are analyzed by subgroups, except for people with the NH hearing condition, for whom it is lower.

The mean tinnitus frequency of our cohort was 5072 Hz, similar to that of Brueggemann et al. (5058 Hz) [12]. Kang et al. [30] found a lower tinnitus frequency (4973 Hz) in patients with age-related hearing losses and a higher tinnitus frequency (5636 Hz) in subjects with noise-induced hearing losses. The mean tinnitus frequency was higher for the male and NH subgroups but lower for the female subgroup. This difference was statistically significant at *p* < 0.05.

The mean THI score of the 270 subjects of our study was 49, higher than those of Van den Berge et al. (42.5) [11] and Aazh and Salvi (45) [29]. This mean value (49) was the same for the male/female and HI/NH subgroups, but different for the compensated/decompensated subgroups, as the THI score is used just to define these subgroups. Mores et al. [31] reported moderate and moderate/mild distress for the HI and NH subgroups, respectively. Savastano et al. [32] reported higher subjective discomfort in HI patients, which supported the association between hearing loss and tinnitus.

Despite the fact that the correlation coefficients between AAT and tinnitus frequency and tinnitus severity were weak (although with high significance levels), two trends should be emphasized: the tinnitus frequency decreases slightly with AAT, and THI increases with AAT. A similar positive correlation between tinnitus severity and hearing losses was reported by Nicolas-Puel et al. [33]. Udupi et al. [34], on the other hand, did not find a significant relationship between hearing status and tinnitus severity.

This study has the limitation of not having considered the psychological, psychiatric, and personality variables of the participants, although they are demonstrated to have a strong influence on the tinnitus-related distress of patients [9,10,11,12,35,36]. Another important limitation is that the hearing testing of participants were performed at different external clinics.

## 5. Conclusions

The heterogeneity of 270 tinnitus participants has been analyzed regarding their gender, hearing condition, and tinnitus severity. Male participants showed statistically significantly higher averaged auditory thresholds than female participants. Moreover, the mean tinnitus frequency was significantly higher in men than women. The mean tinnitus frequency was significantly lower in hearing-impaired than in normal-hearing subjects. Moreover, normal-hearing participants seem to initiate their tinnitus earlier. Regarding tinnitus severity, participants with compensated and decompensated tinnitus presented differences in their laterality (a higher percent towards the left ear in the compensated subgroup, and bilateral in the decompensated subgroup) and tinnitus sound type (a higher percent of tonal in the compensated subgroup and hissing in the decompensated subgroup).

## Figures and Tables

**Figure 1 brainsci-13-00652-f001:**
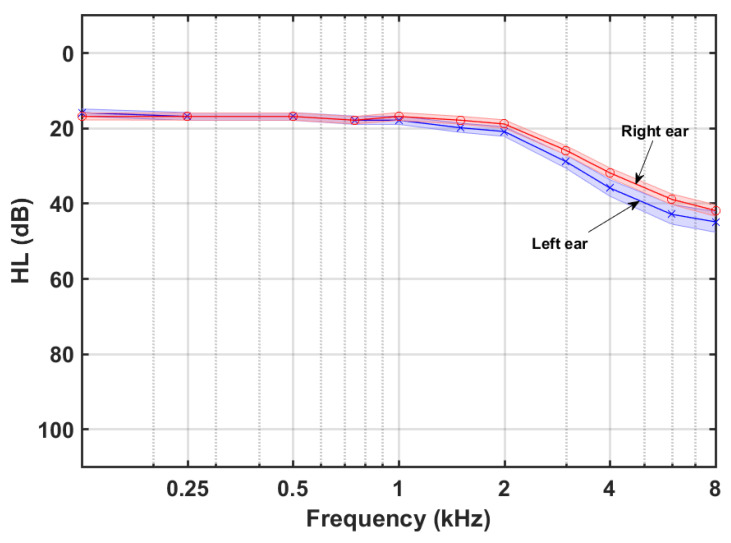
Mean hearing levels of the 270 participants in left and right ears. Shaded area around the HL curves outlines confidence intervals.

**Figure 2 brainsci-13-00652-f002:**
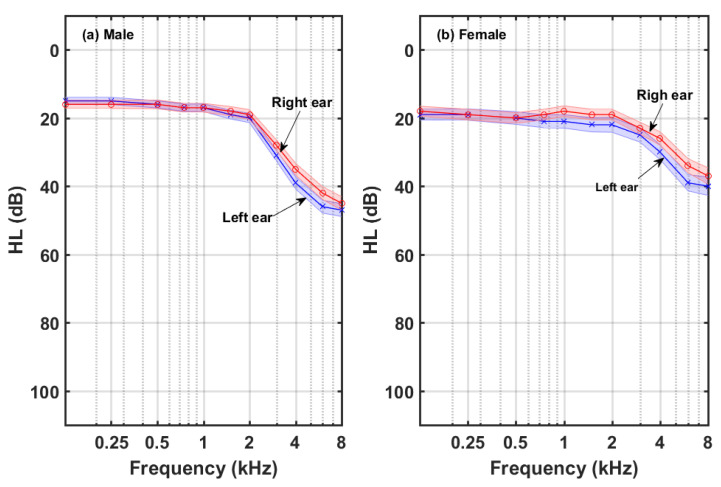
Mean hearing levels of the (**a**) 178 male and (**b**) 92 female participants in left and right ears. Shaded area around the HL curves outlines confidence intervals.

**Figure 3 brainsci-13-00652-f003:**
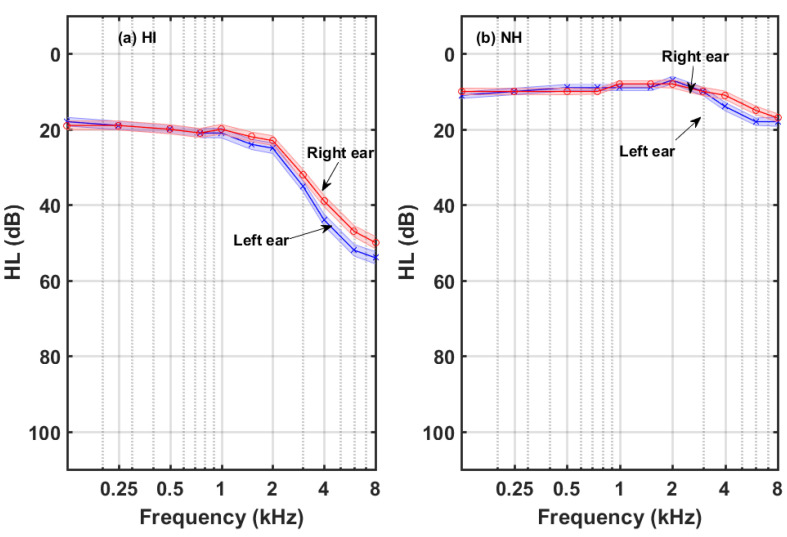
Mean hearing levels of the (**a**) 201 HI and (**b**) 69 NH participants in left and right ears. Shaded area around the HL curves outlines confidence intervals.

**Figure 4 brainsci-13-00652-f004:**
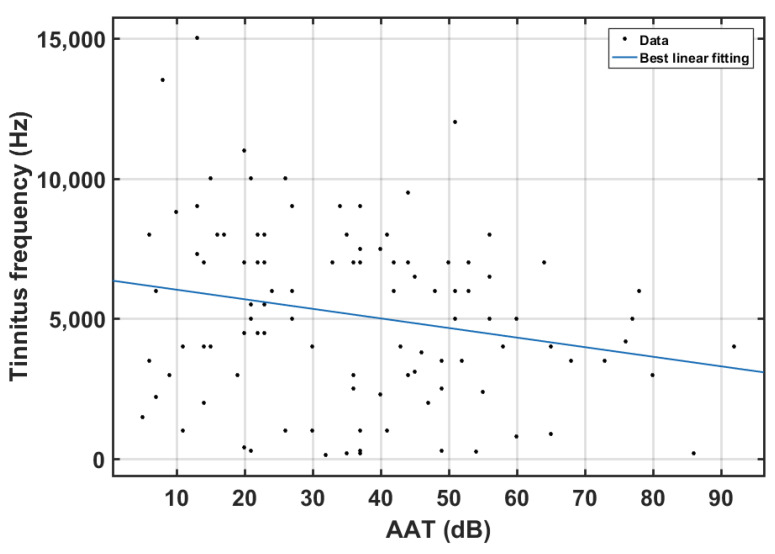
Scatter plot of tinnitus frequency versus AAT for the decompensated subgroup.

**Figure 5 brainsci-13-00652-f005:**
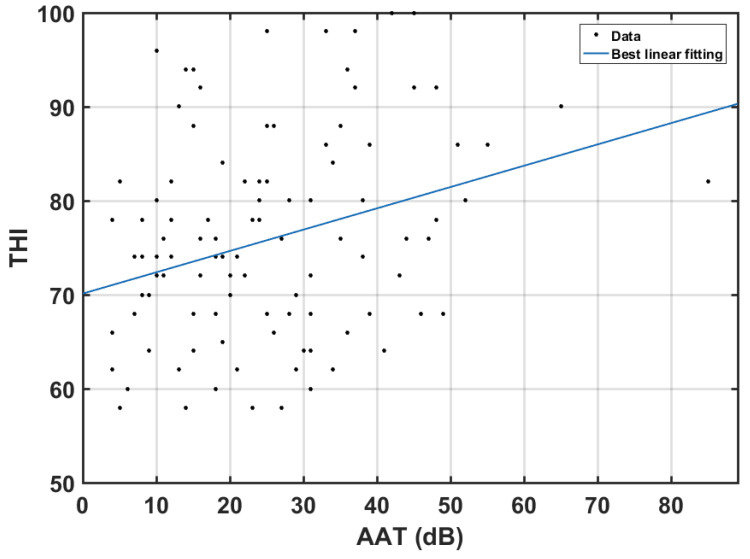
Scatter plot of tinnitus severity versus AAT for the decompensated subgroup.

**Table 1 brainsci-13-00652-t001:** Gender and age of participants.

	All	Male	Female
N	270	178 (66%)	92 (34%)
Age (mean) in years	51.3	51.9	49.9
Age (SD) in years	10.8	10.9	10.8

**Table 2 brainsci-13-00652-t002:** Tinnitus location and tinnitus sound type of participants.

		Tinnitus Location			Tinnitus Sound Type		
		Bilateral	Left Ear	Right Ear	Hissing	Tonal	Ringing
All (270)	N	133	92	45	102	90	78
	%	49	34	17	38	33	29
Male (178)	N	88	61	29	69	62	47
	%	50	34	16	39	35	26
Female (92)	N	45	31	16	33	28	31
	%	49	33	17	30	36	30
HI (201)	N	100	68	33	83	63	55
	%	50	34	16	41	31	28
NH (69)	N	33	24	12	19	27	23
	%	48	35	17	28	39	33
Comp. (165)	N	77	61	27	57	58	50
	%	47	37	16	35	35	30
Decomp. (105)	N	56	31	18	45	32	28
	%	53	30	17	43	30	27

**Table 3 brainsci-13-00652-t003:** Age and tinnitus duration of participants.

	Age (Years)		Tinnitus Duration (Years)	
	Mean	SD	Mean	SD
All (270)	51.3	10.8	6.5	8.8
Male (178)	51.9	10.8	7.3	9.2
Female (92)	49.9	10.8	4.8	7.5
HI (201)	53.0	11.0	7.8	8.8
NH (69)	46.0	7.0	5.0	8.5
Comp. (165)	52.2	11.1	7.5	9.4
Decomp. (105)	49.7	10.2	4.9	7.4

**Table 4 brainsci-13-00652-t004:** Hearing levels, tinnitus frequency, and tinnitus severity of participants.

	Average Auditory Thresholds (dB)						Tinnitus Frequency (Hz)		THI	
	AAT		AAT_LH_		AAT_HF_					
	Mean	SD	Mean	SD	Mean	SD	Mean	SD	Mean	SD
All (278)	25	14	18	13	37	20	5072	2931	49	25
Male (178)	25	14	17	12	39	20	5543	2889	49	25
Female (92)	24	15	20	14	32	18	4163	2811	50	25
HI (201)	29	13	21	13	44	17	4966	2700	50	24
NH (69)	11	6	10	6	15	7	5383	3524	47	26
Comp. (165)	25	15	18	12	37	19	5033	2789	32	14
Decomp. (105)	25	13	19	14	36	20	5134	3154	76	11

**Table 5 brainsci-13-00652-t005:** Correlation coefficients between average auditory threshold, tinnitus frequency, and tinnitus severity. Comp. and decomp. stand for compensated and decompensated tinnitus, respectively.

		Tinnitus Frequency							Tinnitus Severity						
		All	Male	Female	HI	NH	Comp.	Decomp.	All	Male	Female	HI	NH	Comp.	Decomp.
All	AAT	−0.214 ***							0.142 *						
	AAT_LF_	−0.258 ***							0.159 **						
	AAT_HF_	−0.127 *							0.102						
Male	AAT		−0.296 ***							0.067					
	AAT_LF_		−0.2812 ***							0.085					
	AAT_HF_		−0.2575 ***							0.282 **					
Female	AAT			−0.102							0.278 **				
	AAT_LF_			−0.178							0.282 **				
	AAT_HF_			0.003							0.229 *				
HI	AAT				−0.267 ***							0.167 *			
	AAT_LF_				−0.301 ***							0.191 **			
	AAT_HF_				−0.155 *							0.098			
NH	AAT					−0.075							−0.045		
	AAT_LF_					−0.145							−0.08		
	AAT_HF_					0.047							0.016		
Comp.	AAT						−0.138							0.205 **	
	AAT_LF_						−0.194 *							0.210 **	
	AAT_HF_						−0.06							0.176 *	
Deco.	AAT							−0.31 **							0.306 **
	AAT_LF_							−0.333 ***							0.283 **
	AAT_HF_							−0.217 *							0.276 **

* significance *p* < 0.05. ** significance *p* < 0.01. *** significance *p* < 0.0.

**Table 6 brainsci-13-00652-t006:** Summary of the mean categorical and quantitative variables. Significantly higher and lower values are marked with symbols ↑ and ↓. Comp. and decomp. stand for compensated and decompensated tinnitus, respectively.

		All	Gender		Hearing Condition		Tinnitus Severity	
			Male	Female	HI	NH	Comp.	Decomp.
Tinnitus laterality	Bilateral (%)	49	≈	≈	≈	≈	≈	↑
	Left ear (%)	34	≈	≈	≈	≈	↑	↓
	Right ear (%)	17	≈	≈	≈	≈	≈	≈
Tinnitus sound type	Hissing (%)	38	≈	↓	↑	↓	↓	↑
	Tonal (%)	33	≈	↑	≈	↑	↑	↓
	Ringing (%)	29	↓	≈	≈	↑	≈	≈
Average Auditory Threshold	AAT (dB)	25	≈	≈	↑	↓	≈	≈
	AAT_LF_ (dB)	18	≈	≈	↑	↓	≈	≈
	AAT_HF_ (dB)	37	≈	↓	↑	↑	≈	≈
Tinnitus onset age (years)		44.8	≈	≈	≈	↓	≈	≈
Tinnitus frequency (Hz)		5072	↑	↓	≈	↑	≈	≈
THI		49	≈	≈	≈	≈	↓	↑

## Data Availability

Data are unavailable due to privacy or ethical restrictions.

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
