# Peer review of "A Heterogeneous Sample of a Spanish Tinnitus Cohort"

_brainsci, 2023, doi:10.3390/brainsci13040652_

Round 1

Reviewer 1 Report

First, I would like to thank the authors for reading this manuscript. The present article deals with different predictive factors of tinnitus, which I find interesting and clinically significant. However, I have found many issues which should be corrected. Regarding this, see my specific recommendations in the following.

Abstract

Instead of tinnitus sound, I recommend using the term tinnitus sound type. Moreover, I miss some basic information regarding statistical analysis in the abstract.

In the keywords, I recommend including gender and age, as these variables were also examined in detail.

Introduction

The first sentence of the introduction. Instead of risk causes, I find better using risk factors. There are some necessary minor corrections, such as vestibular disorders (or damage), and emotional disorders. Moreover, I recommend including vascular risk factors, one of the most significant tinnitus risk factors.

Generally, I miss a basic introduction to tinnitus. First, explain what tinnitus means. Then, its main types, such as primary/ secondary and objective/subjective groups, should be mentioned, including the following citation: [Mavrogeni P, Maihoub S, Tamás L, Molnár A. Tinnitus characteristics and associated variables on Tinnitus Handicap Inventory among a Hungarian population. J Otol. 2022 Jul;17(3):136-139. doi: 10.1016/j.joto.2022.04.003.]

Lines 27-32. When the others present the examples, it should be indicated that these are examples and other options are possible too. Therefore, in the brackets, I recommend including ‘e.g.’ or ‘for example’. Moreover, previous studies have observed a significant correlation with two main psychiatric symptoms, i.e., anxiety and depression, and tinnitus severity. I recommend mentioning this and citing it accordingly.

[Wallhäusser-Franke E, D'Amelio R, Glauner A, Delb W, Servais JJ, Hörmann K, Repik I. Transition from Acute to Chronic Tinnitus: Predictors for the Development of Chronic Distressing Tinnitus. Front Neurol. 2017 Nov 20;8:605. doi: 10.3389/fneur.2017.00605]

[Molnár A, Mavrogeni P, Tamás L, Maihoub S. Correlation Between Tinnitus Handicap and Depression and Anxiety Scores. Ear Nose Throat J. 2022 Nov 8:1455613221139211. doi: 10.1177/01455613221139211.]

Line 41. Normal audiometry is an incorrect term; please use normal audiometry result/ normal audiograms instead.

Lines 43-45. What the authors present here is really important, as this is a great limitation of audiometry/tinnitus pitch matching used in everyday practice. Therefore, I recommend highlighting that a generally used audiometry in everyday practice measures the 125-8000 Hz frequency spectrum, and this is a limitation of the generally used audiometry. Please also mention that according to the classification, tinnitus related to hearing loss belongs to the primary tinnitus group.

Lines 55-57. Based on which test were tinnitus severity grades detected (reference 6)?

In the methods, I miss the presentation of inclusion criteria.

Please correct the typo in the table caption (and instead of at). Moreover, in the Table, I recommend including the % values, too, because in that way it is easier to follow the results.

Lines 76-81. I find it a significant limitation that the participants’ hearing testing was performed in different external clinics. The outcome of audiometry depends on many factors, i.e., testing environment, the type of audiometer, qualification, etc. Moreover, the authors state that the hearing levels were measured in the 125-8000 Hz frequency range. However, the highest frequency in the Figure is 6000 Hz (8000 Hz is not indicated on the axis).

Lines 84-85. Were there any statistically significant differences between males’ and females’ results?

Figure 1. The subtitle indicates that the frequency values are presented in Hz; however, under the diagram, some values (i.e., 250 and 750) are present in Hz, while others (i.e., 1.5, 3 and 6) are in kHz. Please unify. Furthermore, as previously mentioned, include 8000 Hz too. Please perform the same in the case of Figures 2 and 3 too.

Line 119. I recommend using the term case history instead of anamnesis.

Lines 120-125. Overall, the quality of this section is low. For example, include the exact number and/or % values instead of ‘much participants’. Furthermore, I do not recommend using… in a scientific text. Include the examples instead. Moreover, what the authors include in brackets are possible indications for middle ear surgeries. There are surgeries of the external- and inner ear as well; therefore, please define this more accurately. In addition, cholesteatoma and otosclerosis are possible indications for middle ear surgery; hence, please include, e.g., due to, before them. Also, correct the spelling of otosclerosis. Instead of vestibular troubles, I recommend using the term vestibular disorders/damage. The correct name is not Meniere, but Ménière’s disease or syndrome. Furthermore, dizziness is not a diagnosis, but a symptom. Include examples regarding vestibular disorders instead. Please specify what others mean (i.e., other central causes based on the included examples). Moreover, not Chiari, but Chiari malformation. This section must be significantly improved, as it contains essential information, but the quality of presentation is moderate.

Lines 126-133. Tinnitus pitch-matching and tinnitus intensity measurements were not performed?

Lines 134-137. Regarding the THI, please indicate that it measures 3 dimensions of symptoms (i.e., functional, emotional, and catastrophic). Furthermore, it is not highlighted whether the THI was validated in Spanish or not? In the sentence beginning at line 135, no verb can be found; therefore, in this sentence makes no sense in its current form.

Regarding the statistical analysis, I miss some information, e.g., the names of the used tests, the distribution of the data, etc.

Line 166. HI participants were older (7 years). – How could the authors interpret this result?

In Table 2., instead of tinnitus sound, I recommend using tinnitus sound type.

Lines 181-182. Was this difference defined as statistically significant?

Lines 183-184. The mean tinnitus frequencies seem to be too low. In the discussion the authors state that the mean tinnitus frequency of the whole cohort was 5072 Hz; this frequency seems more accurate. The table also presents mean tinnitus frequency values at approximately 5000 Hz. Furthermore, were the differences statistically significant?

Figure 4. This scatter plot also includes values over 8000 Hz; however, only 125-8000 Hz frequencies were mentioned when audiometry was described. Furthermore, this Figure indicates that tinnitus pitch-matching was used, but this is not indicated in the methods. Please clarify.

Table 5. Please explain the abbreviations in the Table caption. Furthermore, this table resembles a summarisation of different correlation coefficients, but the results are not appropriately interpreted in the text. This should be improved.

Lines 211-212. The first sentence indicates that approximately twice as many males as females were enrolled in this investigation.

Line 217. Instead of age of tinnitus onset, I recommend using the term ‘time since tinnitus onset’.

Table 6. Explain the meaning of the symbols and abbreviations in the table caption, too. Instead of tinnitus type, use ‘tinnitus sound type’, and ‘time since tinnitus onset’.

Line 251. How can it be explained that the mean age in tinnitus peaked (and usually peaks) at approximately 50 years of age? Furthermore, please highlight that the average duration of tinnitus refers to a chronic presentation of symptoms.

Lines 261-262. Were these differences defined as statistically significant in that study?

Line 264. It is not unequivocal whether the last sentence of this paragraph refers to the results of the present investigation or to that of previous ones.  

Lines 265-268. At this section, I miss the discussion of previous studies which analysed the influencing factors of THI results. Please improve.

Limitations – Here, it must be included that the hearing testing performed at different external clinics is a significant limitation. Furthermore, if the THI was not validated in Spanish, this is also a limitation. Additionally, tinnitus intensities were not measured (or at least not included) in this study.

Overall, I find the conclusion section too short, and it does not appropriately highlight the clinical importance of the present investigation. Please try to improve.

I look forward to receiving the revised version of this manuscript.

Reviewer 2 Report

1 - The authors studied the effect of different factors in the evaluation and treatment of tinnitus. Title - Clearly defines the work.

2 - The study proposal of this topic is interesting.

3 - This study agrees with other studies, It contributes with a new orientation in the study of patients with tinnitus.

4 - This study has the advantages of dividing the study evaluation into subgroups according to the different factors that can influence the perception and severity with which patients manage tinnitus. This could be a key point in improving intervention in more complex cases. The results obtained make it possible to highlight the existing differences in the different tests carried out and to understand that the characteristics of tinnitus according to gender can explain the way in which men and women experience this symptom.

5 - The conclusions agree with the investigation carried out and the results obtained

6 - References are adequate and up-to-date.

7 - The figures and images are well organized and didactic for a better understanding of the text.

Round 2

Reviewer 1 Report

Thank you for the revised version of the manuscript; the overall quality of the manuscript has been significantly improved. There are two necessary minor corrections.

Introduction first sentence: Not vestibular risk factors but vascular risk factor

Conclusion: Please remove the numbers in the brackets, these are not necessary in the conclusion, which is only a summarisation of the results. 
